# The involvement of non-governmental organisations in achieving health system goals based on the WHO six building blocks: A scoping review on global evidence

Leila Doshmangir[1], Arman Sanadghol[2], Edris Kakemam [3]*, Reza Majdzadeh[4]

1 Department of Health Policy and Management, Tabriz Health Services Management Research Center, School of Management and Medical Informatics, Tabriz University of Medical Sciences, Tabriz, Iran, 2 Department of Health Policy and Management, School of Management and Medical Informatics, Tabriz University of Medical Sciences, Tabriz, Iran, 3 Non-communicable Diseases Research Center Research Institute for Prevention Non-Communicable Diseases, Qazvin University of Medical Sciences, Qazvin, Iran, 4 School of Health and Social Care, University of Essex, Colchester, United Kingdom

* edriskakemam@gmail.com

**Data Availability Statement:** All relevant data are within the paper and its Supporting Information files.

## Abstract

### Background

Non-governmental organisations (NGOs) have the potential to make a significant contribution to improving health system goals through the provision of resources, health services and community participation. Therefore, this paper examines the role of NGOs in achieving health system goals, based on the six building blocks of a health system framework, and identifies strategies to enhance NGO involvement in achieving health system goals.

### Methods

A scoping systematic review methodology was used to map and synthesise the existing literature on the topic, following the latest JBI six-stage framework. Four databases and one search engine including PubMed, Web of Science (ISI), EMBASE, Scopus and Google Scholar were searched from January 2000 to January 2024. The results were synthesised using a directed content analysis approach, and the findings were categorised according to the dimensions of the six building blocks.

### Results

NGO involvement in health system goals can effectively address gaps in service delivery, strengthen the health workforce, improve health information systems, increase access to essential medicines, mobilise resources and promote good governance. In addition, six key strategies were identified, including joint planning, policy development, capacity building, resource allocation, developing collaboration, and improving the quality of health care, to enhance NGO participation in achieving health system goals.

**Funding:** The author(s) received no specific funding for this work.

## Conclusion

NGOs can play a critical role in achieving health system goals, alongside government and other key health stakeholders. Governments need to use evidence-based policies and interventions to support NGOs to realise their potential in achieving health system goals.

## Introduction

The role of non-governmental organisations (NGOs) as alternative healthcare providers to the state has grown in recent years. NGOs are able to pursue the same goals as the state, but they are less constrained by government inefficiencies and resource limitations [1]. Such organisations are frequently driven by a specific mission or cause and endeavour to enhance society through a variety of means [2]. In the field of healthcare, NGOs may focus on a diverse range of issues, including the provision of healthcare services, research, and advocacy [3]. NGOs contribute a distinctive perspective and expertise to the discourse on healthcare [4]. Such organisations are frequently closely associated with the communities they serve, and thus possess a profound comprehension of the local circumstances and requirements [5]. This allows them to innovate and adapt their strategies in response to emerging health challenges [6] in a way that governmental entities are less able to do [7]. Furthermore, NGOs frequently serve as advocates for marginalised populations, striving to ensure that their voices are heard and their needs are met within the healthcare system [8].

NGOs play a vital role in the achievement of health system goals [9] and ensuring that everyone has access to healthcare [10]. Through their contributions to health system financing, they help bridge the financial gaps [11], reduce inequalities [12], and enhance the overall accessibility [13] and quality of healthcare services [14]. In other words, NGOs are essential in achieving health system goals by addressing gaps, collaborating with government entities, and contributing to the enhancement of various components of healthcare systems [15]. Their efforts in delivering services [16], health workforce [17], health information systems [18], medical products, vaccines, and technologies [19], as well as contributing to health system financing [20], have a significant impact on the overall effectiveness and accessibility of healthcare [21]. Recognizing and supporting the invaluable role of NGOs is crucial for establishing sustainable and inclusive health systems that can effectively meet the needs of communities [22]. By working together, NGOs and governments can build stronger healthcare systems and ensure better health outcomes for all [23]. NGOs play a crucial role in addressing healthcare challenges [24] and since healthcare systems strive to provide equitable [25], accessible [26], and quality care to populations [27], NGOs serve as essential partners in complementing and supporting government, other stakeholders [28], and protecting the rights of marginalized populations [29]. Therefore, their efforts not only complement the work of governments but also offer innovative solutions to complex challenges [30].

Although NGOs are generally regarded as more efficient than other providers, there are shortcomings in the quality of their services. These include a lack of outreach activities, an increased incidence of cold chain failures, and the employment of staff who lack the requisite training. The management of NGOs may be hindered by an intricate organisational structure that they are required to navigate in relation to their parent organisation. Conversely, NGOs may also experience resource constraints and management inefficiencies that are comparable to those observed in government-run providers [31]. They are frequently established in areas where there is a lack of alternative service providers. However, they often fail to coordinate

their activities with the government or with each other. The failure to acknowledge the role of the government can result in NGOs replicating services and wasting resources [32, 33]. The literature identifies a lack of aid coordination and the subsequent fragmentation of health activities in many developing countries. The proliferation of competing organisations that duplicate programme support, create parallel projects, divert health service workers from their routine duties and disrupt planning processes has prompted concern among donors and recipients alike [34]. The mapping of the role of NGOs in achieving the goals of the health system and the development of strategies for improving their involvement may be an essential endeavour in providing evidence that is crucial for evidence-based decision-making, the identification of best practices, and the addressing of the gaps and challenges faced by NGOs. Therefore, the objective of this scoping review is to examine the role of NGOs in achieving health system goals, as defined by the World Health Organization (WHO), and to identify strategies to enhance the participation of NGOs in achieving these goals. By examining each of the six building blocks, namely service delivery, health workforce, health information systems, medical products, vaccines and technologies, and health system financing, we investigate the significant contributions that NGOs make in improving healthcare outcomes and expanding access to essential services. It is crucial to comprehend the significance of NGOs in healthcare in order to facilitate effective collaboration and informed decision-making, thereby ensuring the comprehensive reinforcement of global health systems.

## Method

In view of the principal objective of the review, namely to analyse and map the evidence on the role of NGOs in achieving health system goals using the six building blocks of a health system framework, a scoping review was deemed the most appropriate type of review to provide an initial understanding of the existing landscape. Accordingly, this review was conducted in accordance with the most recent JBI guidance for scoping reviews [35].

This approach allowed us to effectively summarize the findings and identify gaps in existing research. The framework comprises six sequential stages: (1) Formulating the research question, (2) identifying relevant studies, (3)eligibility criteria, (4) screening and selecting evidence, (5) extracting data, and (6) data synthesis. To present the results, we followed the PRISMA-ScR (Preferred Reporting Items for Systematic Reviews and Meta-Analyses Extension for Scoping Reviews) checklist (S1 Table) [36].

### Step one: Identifying the research question

A scoping review is typically initiated with the formulation of one or more research questions. Accordingly, the present scoping review is designed to address the following inquiries:

1. What is the role of non-governmental organisations in achieving the goals of health systems?

2. What potential solutions and strategies could be employed to engage NGOs in achieving the goals of the health systems?

### Step 2. Identifying relevant studies

Four electronic databases and one search engine including PubMed, Web of Science (ISI), EMBASE, Scopus, and Google Scholar were searched from January 2000 to January 2024. The research team identified keywords from the studies and then collaborated with a librarian to develop a search strategy. In January 2024, a literature search was conducted using the terms

"non-governmental organization," "non-state provider," "non-government sector," "health service," "health system," and "health service delivery" in the PubMed, Web of Science (ISI), EMBASE, and Scopus databases, and Google Scholar search engine. The keywords in the peer-reviewed literature search were combined using the Boolean terms "AND" and "OR" in all the explored electronic databases. The S2 Table contains a comprehensive list of the searches carried out in the databases. To ensure comprehensiveness in the literature reviews, the reference lists of included articles were then hand-searched to identify any other potentially relevant articles.

## Step 3. Eligibility criteria

Articles were considered eligible if they met the following criteria: (i) they were published in English, (ii) they were conducted between 2000 and December 31, 2023, (iii) they were published in peer-reviewed journals, (iv) they were either original research or review studies, and (v) they included sufficient data to analyze the impactful interventions of NGOs in health-related activities and the health system. On the other hand, studies were excluded if they (i) were not available as full-text, (ii) were not peer-reviewed and were only published as abstracts, conference proceedings, summaries, letters to the editor, commentaries, or opinion pieces, and (iii) lacked quantitative or qualitative details.

## Step 4. Evidence screening and selection

The retrieved studies were initially imported into Endnote X20 to facilitate the identification and removal of duplicates. Following the retrieval of articles, all instances of duplication were removed manually by the reviewers. Subsequently, two reviewers (LD and AS) undertook an independent screening of titles and abstracts in order to exclude studies deemed irrelevant. Subsequently, the full texts of the remaining articles, following the initial screening, were evaluated for eligibility. In the event of any discrepancies between the two reviewers, a consensus was reached or the decision was confirmed by a third reviewer (EK). In accordance with the scoping review methodology, no quality assessment or risk of bias assessment was conducted for the included studies, as the objective was to rapidly map the evidence [35].

## Step 5. Data extraction

We created a data extraction form. We extracted and organised details of studies such as first author, country, date, type, study design, quality assessment, field of activity, intervention(s), implementation considerations, role of NGOs based on the six building blocks and strategies using Microsoft Excel. Two reviewers (LD and EK) extracted relevant data from the retrieved articles for a narrative synthesis. In case of disagreement between the reviewers during the data extraction process, a third reviewer (RM) was involved to make the final decision.

## Step 6. Data synthesis

The data was synthesised using the Framework Method, in accordance with the six building blocks of the WHO [37]. The Framework Method represents a systematic approach to qualitative content analysis. The method provides a transparent structure for the synthesis of data within the analytical framework, facilitating the identification of patterns and themes while maintaining a comprehensive overview of the entire data set. The WHO framework is comprised of six blocks, including service delivery, the health workforce, health information systems, medical products, vaccines, and technologies, as well as health system financing. Two authors (LD and AS) undertook an independent synthesis of the data, which was analysed

using the framework synthesis method. This was done in order to identify similarities and differences between the literature and the selected model, with a view to establishing which aspects of the studies would map against it. The findings of the studies were subjected to a detailed examination, and the primary codes were extracted. Following the extraction of the initial codes and a review thereof, all roles and strategies extracted from the selected studies were compared and grouped into the six blocks of the selected framework.

## Result

### Selection of evidence sources

The process of reviewing and presenting the search results is shown in Fig 1. Initially, a keyword search yielded 1424 articles. After screening the title and abstracts and retrieving records, 140 studies were selected for a thorough review of the full text. Data extraction and analysis was then performed on 85 studies published between 2000 and 2023.

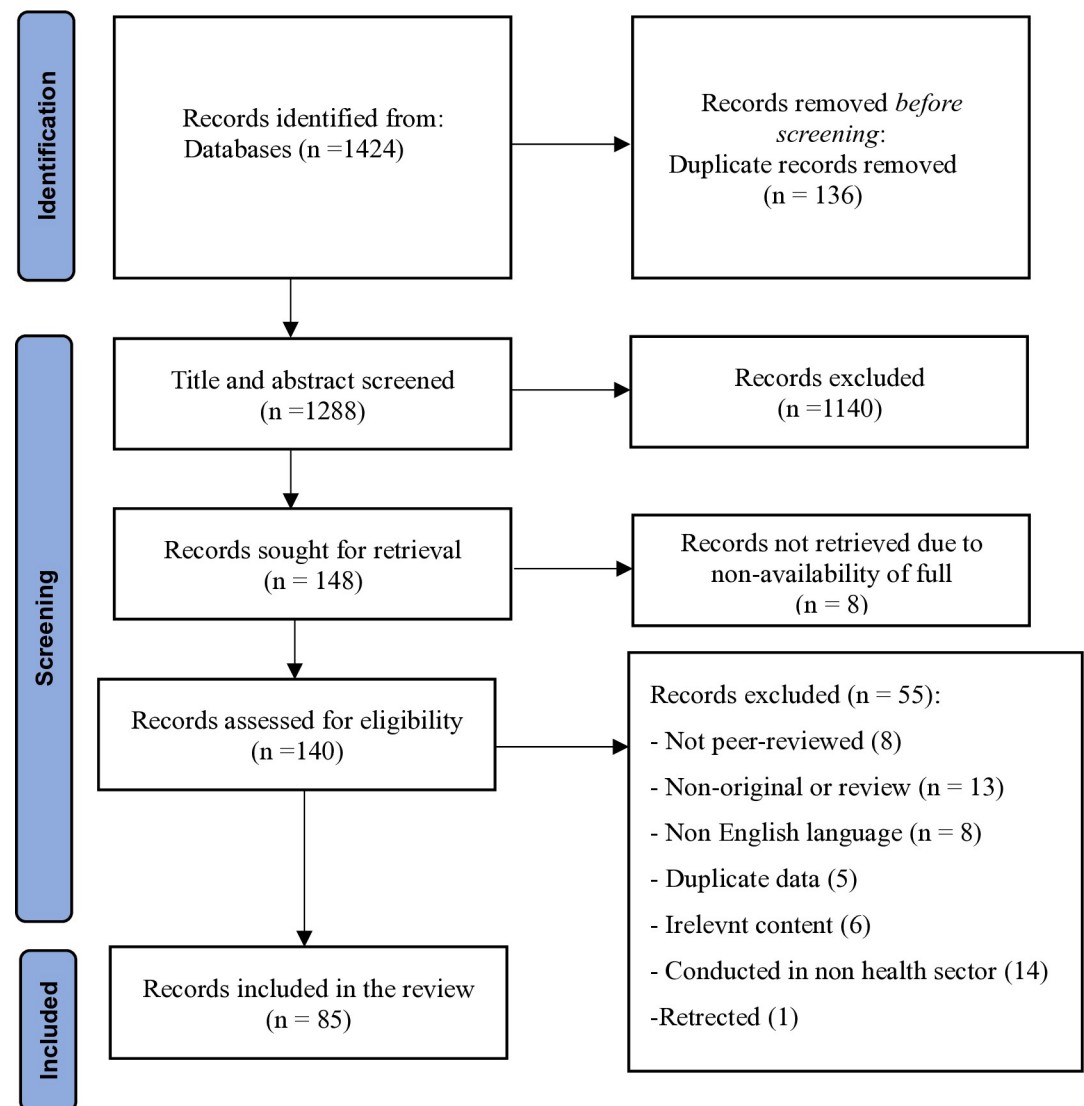

**Fig 1. PRISMA flow diagram.**

## Characteristics of evidence sources

Studies were conducted in 51 countries and five regions (Eastern and Central Europe, Sub-Saharan Africa, South and Southeast Asia, the Caribbean and Latin America). Ten studies were conducted in high-income countries, 20 in upper-middle-income countries, 51 in lower-middle-income countires and 19 in low-income countries (LICs). 73 studies reported that NGOs provided primary care, 4 studies referred to secondary care, and 8 studies referred to tertiary care. Based on the Health Research Classification System (HRCS) [38], there were studies on 8 of the 21 HRCS categories. 32 studies were on general health topics, 24 studies were on infectious diseases, 22 studies were on reproductive health and childbirth, 4 studies were on mental health topics, 3 studies were on cancer and neoplasms, 2 studies were on cardiovascular topics, 3 studies were on inflammation and the immune system, and 1 study was on musculoskeletal topics (See more details in S3 Table).

## Synthesis of results

Based on the six building blocks, eighteen cases of function were identified related to the involvement of NGOs in achieving health system objectives (Table 1).

## Service delivery

One of the factors influencing NGOs' involvement in achieving health system goals is service delivery. NGOs provide services through the provision of health facilities, health care services, and services for specific conditions. 76 studies focused on service delivery by NGOs [39–115], including health care services, services for specific diseases (such as HIV/AIDS, Ebola, tuberculosis, reproductive, and cardiovascular), and health facilities as strategies for NGO involvement.

## Health workforce

Through a variety of strategies and interventions, NGOs provide or support the provision of health services or the operation of health facilities by providing skilled personnel. Fifty-six

**Table 1. NGOs participation based on the six building blocks.**

| Main blocks | | Subscales | Number of studies |
|---|---|---|---|
| Service delivery | | Health care | 29 |
| | | Health facilities | 33 |
| | | Special diseases | 68 |
| Health workforce | | Health care professionals | 37 |
| | | Community Health Worker/ Community | 43 |
| Information | | Health- oriented programs | 35 |
| | | Information support | 1 |
| Medical products, vaccines and technologies | | Medical products | 24 |
| | | Vaccines | 7 |
| | | Equipment and technologies | 6 |
| Financing | Financing source | Free/ subside | 32 |
| | | Insurance | 7 |
| | Financial support | Donation/ Grant | 20 |
| | | International organization | 15 |
| | | Self-Financing | 8 |
| Leadership and governance | | Interact with State | 31 |
| | | Interact with health service providers | 17 |
| | | Interact with citizen | 10 |

studies focused on health workers through NGOs involvement [39, 41–43, 45–47, 50, 52–57, 63–66, 68, 70–75, 79, 80, 82–89, 92–96, 102–105, 109, 111–120]. Of these, 37 refer to the use of health professionals and 43 studies refer specifically to community health workers/ communities.

### Health information systems

NGOs use different sources of information needed to promote public health and raise awareness among the target population. Thirty-three studies focused on the third core component of the building blocks, health information systems [41, 44, 52, 55, 61, 63, 68, 70, 71, 74, 77, 78, 80, 84, 86–89, 92–96, 99, 100, 109, 110, 115–119, 121]. Of these, all 35 studies reported that NGOs provided health-oriented programmes through awareness raising, training, mobile phone use, outreach, counselling and the like, and only one study reported that NGOs provided information.

### Medical products, vaccines and technologies

In many countries, poor people still have limited access to primary health care. NGOs have developed strategies and interventions to improve people's access to health services, especially for the vulnerable, the poor and those with specific diseases. Interventions using medical supplies allow them to treat a wide range of health problems, reducing mortality and morbidity. 27 studies reported that NGOs use medical products, vaccines, and technologies [39–42, 44, 46, 54, 57, 63, 65, 66, 68, 71, 74, 76, 78, 83, 85, 89, 92–95, 98, 99, 109, 112]. Of these studies, about 24 reported that NGOs provided medical products and seven reported that NGOs provided vaccines. Six studies focused on NGOs providing technology to their target population.

### Financing

Protecting people from financial hardship has been achieved by supporting them through a range of strategies, including the provision of free health services, the introduction of insurance schemes or the provision of subsidies. The fifth of the six building blocks of the WHO framework, financing, was the focus of 41 studies [39–44, 46, 48, 49, 52, 53, 55, 56, 58, 59, 62–65, 68–75, 78, 83, 88, 90, 91, 93–95, 103, 109, 111, 112, 114, 122]. The component was divided into two parts, 35 studies focusing on the financial support provided by NGOs to their target population and 39 studies focusing on the sources of NGOs funding. Of these, some 32 studies reported that NGOs provided free or subsidised services and seven studies reported that NGOs provided insurance plans. 20 studies reported that NGOs received donations or grants, 15 studies reported that NGOs received funding from international organisations and 8 studies reported that NGOs were self-financing in various ways.

### Leadership and governance

NGOs build coalitions at different levels, they are at the state level and in contact with health policymakers, with health service providers, and also to be the voice of the people, to be in contact with the community. 42 studies referred to the role of NGOs in leadership and governance [40–42, 44–47, 49, 50, 52, 55, 58, 61, 66, 67, 70, 72, 78, 80–82, 86–89, 92–95, 100, 104, 106, 110–116, 119, 120, 122]. Of these, 31 studies focused on the level of NGOs interaction at the government level, 17 studies at the level of health care providers, and 10 studies at the level of citizens.

### Strategies for involving NGOs in achieving health system goals

Table 2 are presented 6 themes, 16 sub-themes and 43 strategies of NGO participation in achieving health system goals. The strategies identified through the review for improving

**Table 2. Themes, sub-themes, and strategies of NGOs' participation in achieving the goals of the health system.**

| Theme | Sub-theme | Strategies |
|---|---|---|
| **Joint planning** | **Increase access** | • Offering healthcare services/promoting wellness within the community [39–48, 52–66, 68–91, 96, 98, 101–105, 107–110, 112–115, 119, 121, 122]<br>• Providing health facilities [46, 52, 66, 70, 115] |
| | **Increase equity** | • Strengthening the health system governance [52, 70, 95]<br>• Providing fair treatment in healthcare [67, 85, 94, 115] |
| **Policy development** | **Aligning objectives** | • Aligning policies with moving objectives [67, 93] |
| | **Supporting NGOs** | • Supporting the role and importance of NGOs [49, 93, 118]<br>• Accompanying NGOs by government agencies [95]<br>• Identification of priorities by the MOH and outsourcing them to NGOs [93, 97]<br>• Distribution of prevention materials [57]<br>• Getting policymakers and health researchers to recognize the deficiencies and shortcomings of the current NGO collaboration model [97]<br>• The provision of assistance to indigenous NGOs [49, 116] |
| | **Preparing policy tools** | • Defining the roles and responsibilities of NGOs and distinguishing them from other organisations [51, 88, 93, 115]<br>• Defining the nature and schedule of delivery from NGOs for section health [88]<br>• Defining a plan for scaling and sustainable development for NGOs from the government [88, 93]<br>• Ensuring political support and policymakers' trust in NGOs [67] |
| | **Community-based interventions** | • Conducting interventions through community participation [81, 89, 103, 104]<br>• Conducting interventions through CHWs [52, 80, 96, 102] |
| **Capacity building** | **Training and education healthcare workers** | • Considering financial and nonfinancial incentives, especially tangible rewards [67, 80]<br>• Providing clinical education [74]<br>• Training provision for CHWs [80, 96, 117] |
| | **Raising awareness of the community** | • The dissemination of health information and the provision of instruction to members of a community [44, 57, 61, 68, 77, 85, 110]<br>• Enhancing community awareness and providing training [68, 70, 77, 84, 110]<br>• The editing and preparation of materials designed to prevent disease [57]<br>• The distribution of informational materials in accordance with the public health guidelines [57] |
| | **Empowering NGOs** | • NGOs arranging training sessions to enhance knowledge and skills [57]<br>• Implementing creative and culturally relevant interventions [51] |
| **Resource allocation** | **Promoting innovative models and mechanisms** | • Lending to the poor [70]<br>• Utilizing cell phones for patient communication [71]<br>• Establishing care facilities for individuals in need of nursing services [101, 110] |
| | **Expand financial coverage** | • Providing financial support for community health insurance programs within the community [55, 56]<br>• Financial aid for the local population [70, 84, 98, 121] |
| | **Sharing resources** | • Information sharing among stakeholders [106]<br>• Dedication to gather funds [106]<br>• Collaborating on tangible assets [60, 70, 74, 113] |
| **Collaboration development** | **Promotion participation** | • Fostering collaboration among government agencies, NGOs, and the private sector [67, 70, 99, 111]<br>• Enhancing collaboration across sectors [70, 93, 95, 115]<br>• Establishing transparent and accountable structures and mechanisms for coordination [88, 93, 95, 97]<br>• Committing to a lasting partnership [93, 95, 97, 106]<br>• Establishing fresh connections through the adoption and implementation of health interventions supported by evidence [106]<br>• Employing decentralized frameworks to encourage local ownership and sustainability of programs [106] |
| | **Developing communication channels** | • Enhancing the connections and collaborations among NGOs [57, 70, 95] |
| **Improvement quality healthcare** | **Supporting quality assurance measures** | • Develop a system for evaluating the standard of healthcare [93, 116]<br>• Set up and enhance the monitoring and evaluation of health services' quality [106] |

NGOs participation in achieving the goals of the health system included 6 themes: joint planning, policy development, capacity building, resource allocation, developing cooperation, and improving the quality of health care.

## Discussion

We systematically reviewed studies that examined the role of NGOs and their engagement strategies in achieving health system goals based on the six building blocks of the WHO framework. We found that, NGOs are often at the forefront of addressing health system gaps and challenges within the health system [67]. They have the flexibility to identify areas for improvement and implement innovative solutions [123]. For example, in regions with limited access to health services, NGOs can set up mobile clinics or community health centres to reach underserved populations [124]. NGOs can also focus on specific health issues, such as maternal and child health or infectious diseases, and design targeted interventions to address these challenges [125]. For example, an international study revealed that community-based intervention packages implemented by NGOs led to effective in reducing child mortality in diverse settings [89]. In this context, we found that most NGOs aim to expand service delivery, improve financing and provide essential medical products to help meet the health needs of countries' populations [94].

Collaboration between NGOs and government agencies is key to achieving health system goals. While NGOs bring expertise and grassroots understanding, governments have the resources and power to implement systemic change [100]. By working together, NGOs and government agencies can leverage their respective strengths and create a more comprehensive and effective health system [106]. This collaboration can include joint planning, sharing of resources and coordination of efforts to maximise impact and ensure that health system goals are achieved [112]. This finding is consistent with the policy framework on multi-stakeholder partnerships, which emphasizes the importance of inclusive decision-making, joint planning and shared resources to achieve sustainable development [126]. On the other hand, the public-private partnership (PPP) framework allows the government to involve NGOs, the PPP framework allows the government to involve NGOs [127]. This framework uses the resources and skills of NGOs, governments and the private sector to achieve common goals, such as the development of health infrastructure and the provision of social health services [128]. Of course, it's important to note that these frameworks can vary according to the specific political, social and cultural contexts within a country or region [129].

Over the years, significant reforms have been undertaken with regard to NGOs in the health sector [130]. These reforms aim to strengthen the capacity and effectiveness of NGOs in health service delivery, ensure transparency and accountability, and promote collaboration with government and other stakeholders [92]. One of these reforms is strengthening the capacity of NGOs; governments have recognised the importance of increasing the competence of NGOs to manage health services effectively [95]. Capacity-building efforts focus on increasing the skills, supervision, technical expertise and knowledge of NGO staff [131]. Initiatives such as training programmes, seminars and consultancy services are being expanded to improve their ability to design, implement and evaluate health interventions [132]. NGOs have provided competent personnel for the delivery of health services, using a mix of external and internal stimuli; for example, it has been shown that the act of decision-making, the organisation's vision, mission and strategy, and the competencies and skills of NGO personnel have a positive impact on the effectiveness of NGOs in the delivery of health services [112].

On the other hand, reforms related to NGOs in the health system have been implemented by various countries worldwide [92]. These reforms aim to improve the overall effectiveness,

transparency and accountability of NGOs in the delivery of health services [95]. For example, India has one of the largest NGO sectors in the world and plays a significant role in the delivery of health services, particularly in rural and marginalised communities [122]. To address the challenges of unregulated NGOs, the Indian government launched the National Health Mission (NHM) in 2013 [133].

The NHM aims to strengthen the health system, including through better engagement of NGOs [133]. Reforms introduced under the NHM include NGO accreditation, results-based funding, and regular monitoring and evaluation of NGO activities [134]. In Kenya, health system reforms have been implemented to increase the effectiveness and streamline the work of NGOs [135]. Key reforms include the creation of a legal framework to regulate NGO activities, the establishment of a partnership and coordination framework with NGOs, and performance-based funding [136]. In Australia, the government has introduced a competitive grant system whereby NGOs must demonstrate their ability to deliver effective and evidence-based health services in order to receive funding. This ensures rigorous scrutiny and quality control before funds are allocated [125]. These examples show how different countries have introduced reforms to strengthen the role and effectiveness of NGOs in the health system. These reforms increase accountability, promote coordination and ensure better health outcomes for the target populations.

Governments have established mechanisms to provide financial support to NGOs working in the health sector [50]. These include funding opportunities through grants, contracts and subsidies [131]. Financial reforms aim to improve the availability of resources to NGOs, enabling them to deliver health services effectively, invest in infrastructure and recruit skilled staff [93]. One of the mechanisms used by NGOs in collaboration with government in recent years is social health insurance (SHI). Social health insurance schemes are typically structured so that individuals make financial contributions to a centralised fund [137, 138]. These contributions can be made either indirectly, through taxes, or directly, through wage-based payments [139]. However, because of the financial constraints faced by certain groups of the population, many countries have introduced hybrid SHI systems [140, 141].

NGOs contribute to the efficacy of health systems by measuring the outcomes of programmes and providing support for rational planning. While health information systems in lower-income countries may be inadequate, NGOs frequently invest significantly in the development of robust data collection systems in order to meet the requirements of donors [142]. NGOs often use health information to tailor their programmes, allocate resources effectively and respond to the specific needs of communities [143]. In Ghana, for example, NGOs such as the Ghana Health Service and the Navrongo Health Research Centre rely heavily on health information to improve health services. These organisations collect data on disease prevalence, mortality and health infrastructure to identify priority areas [144]. In India, NGOs such as Pratham and PATH have used health information to address specific issues [145]. Health data helps NGOs in India identify prevalent diseases such as tuberculosis, malnutrition or cataracts and design appropriate interventions [145]. In both Ghana and India, NGOs work with local governments and international health organisations to collect and analyse data to ensure that their programmes are aligned with national health policies and goals. However, there are potential challenges to the use of health information by NGOs [92]. Some NGOs may have difficulty accessing reliable and up-to-date data in resource-limited settings [106]. There may also be concerns about data privacy, security and sharing between NGOs and government agencies [146].

NGOs' use of health governance and leadership is crucial in addressing health challenges in many countries [147]. Through advocacy, service delivery and capacity building, these organisations contribute to improving health systems and health outcomes [148]. Examples from

countries such as India and Kenya illustrate the important role of NGOs in health governance and leadership. In India, NGOs such as the Indian Health Organization (IHO) work with the government to address various health challenges, including maternal and child health, HIV prevention and rural health services [149]. Through their strong governance structures and leadership, these NGOs play a key role in advocating for policy changes and implementing innovative strategies to improve health outcomes [149]. In Kenya, NGOs such as the Kenya Red Cross Society and AMREF Health Africa actively participate in health governance by engaging with policy-makers, advocating for policy change and mobilising resources for health initiatives [150, 151]. These organisations also provide leadership by implementing community-based health solutions and training health workers, thereby strengthening the health system at the grassroots level [150, 151]. It is important to note, however, that the effectiveness of health governance and leadership by NGOs depends on several factors, including the availability of funding, government support, and community commitment [152].

## Limitations

It is important to consider the limitations of the study. A comprehensive literature search was conducted in the four major electronic databases. However, no other databases were searched, nor was the 'grey' literature considered. It is therefore possible that additional relevant studies may have been overlooked. Secondly, in accordance with the methodology employed for the review of the literature, the potential for bias was not evaluated in the studies included in the review. However, we adhered to a rigorous protocol for scoping reviews, which included regular search and extraction meetings to ensure consistent adherence to the inclusion/exclusion criteria and results synthesis process across all team members. It is therefore recommended that caution be exercised when drawing conclusions based on the combined data from these studies.

## Conclusion

This analysis has highlighted the ability of NGOs to play a critical role in achieving health system goals, in collaboration with government and other key health sector actors. These organisations have been actively involved in service delivery, health workforce development, information management, procurement of medical products, vaccines and technologies, and leadership and governance. NGOs and governments can work together to build partnerships, establish effective channels of communication and align their goals. This collaboration can include joint planning, policy development, capacity building and resource allocation to achieve common health system goals and ensure equitable access to health services. By working together, NGOs and governments can leverage their strengths, pool resources and coordinate efforts to build more comprehensive and sustainable health systems. The involvement of NGOs in the pursuit of health system goals based on the six essential building blocks is essential. Their involvement can address gaps in service delivery, strengthen the health workforce, improve health information systems, facilitate access to essential medicines, mobilise resources and promote good governance. Governments can benefit from the expertise and resources of NGOs by partnering with them to achieve better health outcomes for all.

## Supporting information

**S1 Table. PRISMA 2020 checklist.**
(DOCX)

**S2 Table. Search strategies.**
(DOCX)

**S3 Table. Characteristics of included studies.**
(DOCX)

## Author Contributions

**Conceptualization:** Leila Doshmangir, Arman Sanadghol, Edris Kakemam, Reza Majdzadeh.

**Data curation:** Arman Sanadghol.

**Formal analysis:** Leila Doshmangir, Arman Sanadghol.

**Investigation:** Leila Doshmangir, Arman Sanadghol, Edris Kakemam, Reza Majdzadeh.

**Methodology:** Leila Doshmangir, Arman Sanadghol, Edris Kakemam.

**Software:** Arman Sanadghol.

**Supervision:** Leila Doshmangir, Reza Majdzadeh.

**Validation:** Arman Sanadghol, Edris Kakemam.

**Writing – original draft:** Leila Doshmangir, Arman Sanadghol, Edris Kakemam.

**Writing – review & editing:** Leila Doshmangir, Edris Kakemam, Reza Majdzadeh.

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
