## [Decision Letter · Decision Letter 0]

15 Oct 2024

PONE-D-24-03861The involvement of nongovernmental organisations in achieving health system goals based on the WHO six building blocks: A scoping review

Dear Dr. Kakemam,

Thank you for submitting your manuscript to PLOS ONE. After careful consideration, we feel that it has merit but does not fully meet PLOS ONE’s publication criteria as it currently stands. Therefore, we invite you to submit a revised version of the manuscript that addresses the points raised during the review process. Please address all comments raised by the reviewers point by point. Any comments you do not adress you must provide a convincing justifcation as to why you could not address it

Please submit your revised manuscript by Nov 29 2024 11:59PM.  If you will need more time than this to complete your revisions, please reply to this message or contact the journal office at plosone@plos.org. Please include the following items when submitting your revised manuscript:A rebuttal letter that responds to each point raised by the academic editor and reviewer(s). You should upload this letter as a separate file labeled 'Response to Reviewers'.A marked-up copy of your manuscript that highlights changes made to the original version. You should upload this as a separate file labeled 'Revised Manuscript with Track Changes'.An unmarked version of your revised paper without tracked changes. You should upload this as a separate file labeled 'Manuscript'.

We look forward to receiving your revised manuscript.

Kind regards,

Josue Mbonigaba, Ph.D

Academic Editor

PLOS ONE

2. Please ensure that you include a title page within your main document. You should list all authors and all affiliations as per our author instructions and clearly indicate the corresponding author.

Reviewers' comments:

Reviewer's Responses to Questions

**Comments to the Author**

1. Is the manuscript technically sound, and do the data support the conclusions?

Reviewer #1: Partly

Reviewer #2: Yes

2. Has the statistical analysis been performed appropriately and rigorously? 

Reviewer #1: N/A

Reviewer #2: Yes

3. Have the authors made all data underlying the findings in their manuscript fully available?

Reviewer #1: No

Reviewer #2: Yes

4. Is the manuscript presented in an intelligible fashion and written in standard English?

Reviewer #1: Yes

Reviewer #2: Yes

5. Review Comments to the Author

Reviewer #1: Topic: The Involvement of NGOs Operating in Iran? or where?

Introduction:

In general, I agree with the presentation. However, also note that: (1) NGOs, Government and Non-state actors, all start from their established interests and positions, and the conclusion you made may not satisfy diverse groups (political, business, advocates, etc.). Therefore, criticisms to government and NGO positions is needed for the author to ensure neutrality. (2) access to and benefit from health care services by the poor and other diverse groups is both right and privilege, I believe and argue. Thus, a balanced stance, and a kind of disaggregated evidence is needed to ascribe "critical ... etc." place to NGOs. (3) A push should be on government to provide health services to all, and NGOs should play a limited aspect of it, via maybe, partnership, and yet, a working partnership between government and NGOs is a contentious area in poor countries where democratic governance barely exist. I suggest the author position the arguments and substantiate claims from the side of all actors (I mean, authors dwell on positive claims, and silent on critiques). There are repetitions, and the research gaps and inquiry are not clearly presented.

Methods

Screening and selection – (1) this an important part of screening and extracting dependable evidence. I am not convinced. Also, the screening steps: (1) using key words, (2) quickly skimming titles, (3) reading abstracts, (4) reading the whole article/document, and (5) clearing for systematic scoping review. Based on these steps, how many articles/documents were generated and how many of them were dropped in each step, with justifications are needed in this section. (2) Assuming you obtained thousands of articles and documents, and some of them might be duplicate. How did the authors manage duplication; for instance, did the authors use EndNote or Mendeley, or what? (3) data analysis is without theory or model to serve as an analytical lens of interpretation. Since this is scientific piece of work, the data analysis needs a theoretical/analytical lens. Otherwise, the scoping review report may tend to become something of journalistic work. (4) The records screed is 1288, yet the records you excluded are 2159. How does the calculation work? Also, why are these records excluded?

Discussion

A lot of evidence from policy/program/project outcome evaluations show that collaboration, program coherence and partnership related objectives are often from those unmet (unachieved)? Besides, these issues are often on paper, and on the ground, not materialized. Thus, I expect the scoping review results to bring concrete evidence on this (including lessons and best practices, if any). Remember my comment on the issue of positioning by government and NGO actors, and how these actors justify their involvement in the health care system (both claim they are effective and serving the people. Yes, how would you show us, maybe, presenting disaggregated evidence along the six building blocks).

Limitations

The limitations described are bigger than the solutions undertaken. Such presentation is often counter-productive and questions the findings and the discussions made in the body. Tune down!!

Reviewer #2: Title: The involvement of non-governmental organisations in achieving health system goals based on the WHO six building blocks: A scoping review

General Overview:

The manuscript presents a scoping review on the involvement of non-governmental organizations (NGOs) in achieving health system goals, using the WHO’s six building blocks framework as a foundation. The study synthesizes literature from various databases and outlines key strategies that NGOs use to contribute to health system improvements. This review is timely and addresses a significant gap in knowledge regarding the role of NGOs in health system strengthening, particularly in low- and middle-income countries.

Strengths:

Relevance and Importance: The involvement of NGOs in health system development is a crucial topic, particularly in regions where government resources are limited. The manuscript addresses the diverse ways NGOs contribute to health service delivery, workforce strengthening, and governance, making it highly relevant for policymakers, healthcare planners, and stakeholders in global health.

Comprehensive Review Methodology: The use of a scoping review methodology following the JBI framework ensures a broad mapping of available evidence. The review includes multiple databases and uses clear inclusion/exclusion criteria, which enhances the credibility of the findings. The synthesis of results according to the six building blocks is a logical and structured approach, making the data easy to follow and understand.

Structured Presentation: The division of results into the six building blocks, along with clear tables outlining strategies and sub-themes, is well-organized. This presentation allows readers to grasp the complexity of NGO involvement in health system strengthening effectively.

Areas for Improvement:

Lack of Quality Assessment: Although the scoping review methodology does not typically include a formal risk of bias assessment, the manuscript would benefit from some discussion of the quality of the included studies. Since the review covers a wide range of study designs, acknowledging potential limitations in the robustness of the included evidence could add transparency to the findings. Including a brief quality assessment, even if informal, would provide more depth to the discussion.

Over-reliance on Qualitative Data: The study synthesizes both qualitative and quantitative data, but the emphasis seems to be on qualitative outcomes, particularly in describing NGO strategies and roles. While this is valuable, there is a lack of quantitative analysis that could further strengthen the conclusions, especially regarding the effectiveness of specific interventions. Including more data on the impact of NGO involvement, in terms of health outcomes or system efficiency, would make the findings more compelling.

Vagueness in Some Strategies: While the manuscript identifies several strategies used by NGOs, some of these are too broad or vaguely described. For instance, terms like "strengthening intersectoral coordination" or "promoting links between NGOs" lack specificity in terms of practical implementation. It would be beneficial for the authors to provide more detailed examples or case studies to illustrate how these strategies have been effectively employed in real-world settings.

Geographical Limitations: The review covers a wide range of studies but would benefit from a more explicit discussion of the geographical distribution of the findings. Are certain regions or countries more represented than others? This could introduce biases in the generalizability of the conclusions. A discussion of regional variations in NGO roles or the challenges faced by NGOs in different health systems would enhance the manuscript's depth.

Clarification of Scope: The manuscript occasionally overlaps in its discussion of NGOs versus other private-sector entities. It would be helpful to clearly delineate the distinction between non-profit, NGO, and private-sector involvement in the health system. This clarification would provide more focus and avoid conflating different types of health system actors.

Methodological Clarifications:

The search strategy is well-documented, but it would be beneficial to explain the choice of 2000 as the starting date for the literature search. Given that NGO involvement in health systems has a longer history, some older studies might still hold relevance.

While the review does not conduct a formal quality assessment, a brief explanation of how study limitations were considered in the synthesis (beyond exclusion criteria) could be helpful for transparency.

Conclusion and Recommendations:

The manuscript provides valuable insights into the role of NGOs in strengthening health systems, but the presentation of the findings could be improved with more specific examples, clearer geographical breakdowns, and attention to the quality of the studies included. The discussion is thorough but could be enhanced by focusing on concrete outcomes or quantitative data showing the impact of NGO involvement. Despite these limitations, the review makes a significant contribution to the understanding of NGO roles in health system strengthening and offers actionable strategies for stakeholders looking to enhance NGO involvement.

6. PLOS authors have the option to publish the peer review history of their article (what does this mean?). If published, this will include your full peer review and any attached files.

Reviewer #1: No

Reviewer #2: No

---

## [Author Response · Author response to Decision Letter 0]

26 Nov 2024

Plos one 

25 November 2024 

Dear Editor, 

Thank you for the comments from you and respected reviewers, which provided us with the opportunity to improve our manuscript. We are pleased to inform you that the authors have had a detailed look at the manuscript and have addressed all comments, as much as possible. The changes to the manuscript were highlighted with the track changes service.

We would hope that you will find these changes up to your satisfaction and look forward to hearing your decision in due course. Please do not hesitate to contact me if any further clarification is required. 

Yours truly,

Corresponding author and on behalf of all co-authors, 

Reviewer 1-

Topic: The Involvement of NGOs Operating in Iran? or where?

Response: This study was conducted in the world literature. 

“The involvement of nongovernmental organisations in achieving health system goals based on the WHO six building blocks: A scoping review on global evidence”

Introduction:

In general, I agree with the presentation. 

However, also note that: (1) NGOs, Government and Non-state actors, all start from their established interests and positions, and the conclusion you made may not satisfy diverse groups (political, business, advocates, etc.). Therefore, criticisms of government and NGO positions are needed for the author to ensure neutrality. 

(2) access to and benefit from health care services by the poor and other diverse groups is both right and privilege, I believe and argue. Thus, a balanced stance, and a kind of disaggregated evidence is needed to ascribe "critical ... etc." place to NGOs. 

(3) A push should be on the government to provide health services to all, and NGOs should play a limited aspect of it, via maybe, partnership, and yet, a working partnership between government and NGOs is a contentious area in poor countries where democratic governance barely exist. 

I suggest the author position the arguments and substantiate claims from the side of all actors (I mean, authors dwell on positive claims, and silent on critiques). There are repetitions, and the research gaps and inquiry are not presented.

Response: Thank you for your valuable comments. The introduction, discussion and conclusion parts were revised based on your comments. 

Methods

Screening and selection – 

(1) this is an important part of screening and extracting dependable evidence. I am not convinced. 

Also, the screening steps: 

(1) using keywords, (2) quickly skimming titles, (3) reading abstracts, (4) reading the whole article/document, and (5) clearing for a systematic scoping review. 

Response: Thank you for your comment. The screening process was revised. 

 Based on these steps, how many articles/documents were generated and how many of them were dropped in each step, with justifications are needed in this section. 

Response: Thank you for your comment. The reasons and justifications were provided for deleting studies.

(2) Assuming you obtained thousands of articles and documents, and some of them might be duplicates. How did the authors manage duplication; for instance, did the authors use EndNote or Mendeley, or what? 

Response: We used the Endnote to manage the retrieved studies. 

(3) Data analysis is without theory or model to serve as an analytical lens of interpretation. Since this is a scientific piece of work, the data analysis needs a theoretical/analytical lens. Otherwise, the scoping review report may tend to become something of journalistic work. 

Response: Thank you for your comment. The WHO six building blocks has been used as a model for data analysis. 

(4) The records screed is 1288, yet the records you excluded are 2159. How does the calculation work? Also, why are these records excluded?

Response: Thank you very much for your valuable comment. The numbers were corrected. 

Discussion

A lot of evidence from policy/program/project outcome evaluations shows that collaboration, program coherence and partnership-related objectives are often from those unmet (unachieved)? Besides, these issues are often on paper, and on the ground, not materialized. Thus, I expect the scoping review results to bring concrete evidence on this (including lessons and best practices, if any). Remember my comment on the issue of positioning by government and NGO actors, and how these actors justify their involvement in the health care system (both claim they are effective and serving the people. Yes, how would you show us, maybe, presenting disaggregated evidence along the six building blocks).

Response: Thank you for your comment. The discussion part was revised based on the comment. 

Limitations

The limitations described are bigger than the solutions undertaken. Such a presentation is often counter-productive and questions the findings and the discussions made in the body. Tune down!!

Response: Thank you for your comment. The limitation part was revised based on the comment. 

Reviewer #2: 

Title: The Involvement of non-governmental organisations in Achieving Health System Goals Based on the WHO Six Building Blocks: A Scoping Review

General Overview:

The manuscript presents a scoping review on the involvement of non-governmental organizations (NGOs) in achieving health system goals, using the WHO’s six building blocks framework as a foundation. The study synthesizes literature from various databases and outlines key strategies that NGOs use to contribute to health system improvements. This review is timely and addresses a significant gap in knowledge regarding the role of NGOs in health system strengthening, particularly in low- and middle-income countries.

Response: Thank you for reading and commenting on our manuscript. 

Strengths:

Relevance and Importance: The involvement of NGOs in health system development is a crucial topic, particularly in regions where government resources are limited. The manuscript addresses the diverse ways NGOs contribute to health service delivery, workforce strengthening, and governance, making it highly relevant for policymakers, healthcare planners, and stakeholders in global health.

Response: Thank you for reading and commenting on our manuscript. 

Comprehensive Review Methodology: The use of a scoping review methodology following the JBI framework ensures a broad mapping of available evidence. The review includes multiple databases and uses clear inclusion/exclusion criteria, which enhances the credibility of the findings. The synthesis of results according to the six building blocks is a logical and structured approach, making the data easy to follow and understand.

Response: Thank you for reading and commenting on our manuscript. 

Structured Presentation: The division of results into the six building blocks, along with clear tables outlining strategies and sub-themes, is well-organized. This presentation allows readers to grasp the complexity of NGO involvement in health system strengthening effectively.

Response: Thank you for reading and commenting on our manuscript. 

Areas for Improvement:

Lack of Quality Assessment: Although the scoping review methodology does not typically include a formal risk of bias assessment, the manuscript would benefit from some discussion of the quality of the included studies. Since the review covers a wide range of study designs, acknowledging potential limitations in the robustness of the included evidence could add transparency to the findings. Including a brief quality assessment, even if informal, would provide more depth to the discussion.

Response: Thank you for the comment. We considered it as a limitation and explained how we considered studies in the synthesis.

Over-reliance on Qualitative Data: The study synthesizes both qualitative and quantitative data, but the emphasis seems to be on qualitative outcomes, particularly in describing NGO strategies and roles. While this is valuable, there is a lack of quantitative analysis that could further strengthen the conclusions, especially regarding the effectiveness of specific interventions. Including more data on the impact of NGO involvement, in terms of health outcomes or system efficiency, would make the findings more compelling.

Response: Thank you for your comment. Actually, due to the nature of our study (scoping review), we have focused on the role of NGOs in achieving health system goals and strategies to enhance NGO involvement in achieving health system goals. Assessing the effectiveness and outcomes of specific interventions is therefore outside the scope of our study.

Vagueness in Some Strategies: While the manuscript identifies several strategies used by NGOs, some of these are too broad or vaguely described. For instance, terms like "strengthening intersectoral coordination" or "promoting links between NGOs" lack specificity in terms of practical implementation. It would be beneficial for the authors to provide more detailed examples or case studies to illustrate how these strategies have been effectively employed in real-world settings.

Response: Thank for your comment. The strategies were revised based on the comment. 

Geographical Limitations: The review covers a wide range of studies but would benefit from a more explicit discussion of the geographical distribution of the findings. Are certain regions or countries more represented than others? This could introduce biases in the generalizability of the conclusions. A discussion of regional variations in NGO roles or the challenges faced by NGOs in different health systems would enhance the manuscript's depth.

Response: Thank you for your comment. The presentation of the findings were revised based on the comment. 

Clarification of Scope: The manuscript occasionally overlaps in its discussion of NGOs versus other private-sector entities. It would be helpful to clearly delineate the distinction between non-profit, NGO, and private-sector involvement in the health system. This clarification would provide more focus and avoid conflating different types of health system actors.

Response: Thank you for your comment. Was revised. 

Methodological Clarifications:

The search strategy is well-documented, but it would be beneficial to explain the choice of 2000 as the starting date for the literature search. Given that NGO involvement in health systems has a longer history, some older studies might still hold relevance.

Response: Based on the comment, we considered the search without a time limit. However, no relevant studies were found before 2000.

While the review does not conduct a formal quality assessment, a brief explanation of how study limitations were considered in the synthesis (beyond exclusion criteria) could be helpful for transparency.

Response: Thank you for your comment. We considered it as a limitation and explained how we considered studies in the synthesis.

Conclusion and Recommendations:

The manuscript provides valuable insights into the role of NGOs in strengthening health systems, but the presentation of the findings could be improved with more specific examples, clearer geographical breakdowns and attention to the quality of the studies included. The discussion is thorough but could be enhanced by focusing on concrete outcomes or quantitative data showing the impact of NGO involvement. Despite these limitations, the review makes a significant contribution to the understanding of NGO roles in health system strengthening and offers actionable strategies for stakeholders looking to enhance NGO involvement.

Response: Thank you for your comment. Was done.

---

## [Editor Report · Decision Letter 1]

28 Nov 2024

The involvement of non-governmental organisations in achieving health system goals based on the WHO six building blocks: A scoping review on global evidence

PONE-D-24-03861R1

Dear Authors 

We’re pleased to inform you that your manuscript has been judged scientifically suitable for publication and will be formally accepted for publication once it meets all outstanding technical requirements.

Kind regards,

Josue Mbonigaba, Ph.D

Academic Editor

PLOS ONE
---

## [Editor Report · Acceptance letter]

20 Jan 2025

PONE-D-24-03861R1 

PLOS ONE

Dear Dr. Kakemam, 

I'm pleased to inform you that your manuscript has been deemed suitable for publication in PLOS ONE. Congratulations! Your manuscript is now being handed over to our production team.

Kind regards, 

on behalf of

Dr. Josue Mbonigaba 

Academic Editor

PLOS ONE